

# Macro and microplastic pollution in Romania: addressing knowledge gaps and potential solutions under the circular economy framework

Florin-Constantin Mihai[1], Simona-Roxana Ulman[1] and Valeria Pop[2,3]

[1] CERNESIM Environmental Research Center, Department of Exact Sciences and Natural Sciences, Institute of Interdisciplinary Research, Alexandru Ioan Cuza University of Iasi, Iaşi, Romania

[2] Research Institute for Sustainability and Disaster Management Based on High-Performance Computing, Faculty of Environmental Science and Engineering, Babes-Bolyai University of Cluj-Napoca, Cluj-Napoca, Romania

[3] Doctoral School "Environmental Science", Babes-Bolyai University of Cluj-Napoca, Cluj-Napoca, Romania

Corresponding author
Florin-Constantin Mihai,
mihai.florinconstantin@gmail.com

## ABSTRACT

**Background:** This review reveals the role of linear economy prevalence and mismanagement practices in plastic pollution of aquatic and terrestrial environments and related knowledge gaps in Romania while outlining downstream and upstream solutions to reduce plastic pollution and adopt circular economy strategies. Thus, the major aim of this study is the investigation of the stage of scientific knowledge concerning all these demands in the Romanian context.

**Methodology:** This work integrates two main approaches: (i) a bibliometric analysis fed by Web of Science and Scopus databases to reveal the current coverage of peer-reviewed literature related to plastic waste in Romania and (ii) a subject-based review to underline the main themes related to plastic waste management, plastic pollution, and mitigating options in Romania in line with circular economy principles.

**Results:** Reducing plastic pollution requires scientific knowledge, multi-sectoral cooperation, and societal awareness. Following this, the topics of plastic waste and plastic pollution appeared to be under-investigated in the literature considering Romania as a case study and concentrated around the 2020 year, emphasizing, in this way, the trendiness of plastic waste concerns and their management in the current research landscape. Our analysis points out that: (i) Romania is facing massive plastic pollution requiring solid improvements in waste management performances; (ii) few peer-reviewed research studies are performed in Romania for both macro and microplastic concerns with unknown pollution levels in most of its geographical regions; (iii) the plastic waste management is still understudied here, while waste statistics are poorly available at local levels; (iv) the perspectives of circular economy transition are still limited, feeding the plastic pollution in the coming years.

**Conclusions:** Several knowledge gaps are identified and must be covered by future research such as (i) adjusting mismanaged plastic waste levels to regional waste management performances and determining littering rates in urban and rural areas to improve the plastic pollution modeling inputs; (ii) examining plastic pollution

associated with landfill sites and waste imports; (iii) assessing the sectoral contributions to macro and microplastic pollution of aquatic environments related to municipalities, tourist destinations, agriculture, *etc.*; (iv) determining retention levels of plastic in river basins and role of riparian vegetation; (v) analyzing microplastics presence in all types of freshwater environments and interlinkage between macroplastic fragmentation and microplastic; (vi) assessing the plastic loads of transboundary rivers related to mismanagement practices; (vii) determining concentrations of microplastics in air, soil, and other land use ecosystems.

# INTRODUCTION

The global increasing waste flows exercise great pressure on the waste management sector, with high concern on the mismanaged plastic waste (MPW), fed by a plastic-driven consumption society following the linear economy model and contaminating every ecosystem on the planet (*Lebreton & Andrady, 2019*; *Winterstetter et al., 2023*). In the same way, in the case of Romania, the incorrect and inefficient waste management, associated with the prevailing consumption model of plastic items that generate a significant quantity of plastic waste ending up in the environment represent challenges in need of solutions (*Mihai & Ulman, 2022*). In detail, Romania is still a landfill-based country (*Romanian Court of Accounts, 2022*). It also registers vulnerability concerning the openness of its citizens to circular initiatives such as product repair, plastic material and/or single-use packaging avoidance, or separate collection of waste (*Ministry of Environment, 2022*). In addition, exposure to illegal plastic waste trade activities (*Mihai & Ulman, 2024*) is present here. Nearby them, institutional challenges to develop policies for a complex cross-sectoral issue, market barriers for recycled products, companies' abilities to grasp opportunities, and good indicators and targets (*EEA, 2022*, p. 13) could be pointed out as other significant barriers in this national context. Consequently, Romania is among the EU countries registering the lowest performances regarding waste management, especially in the matter of waste generation related to economic activity (GDP), waste treatment, and use of recycled materials in the economy (*Ministry of Environment, 2022*, p. 12), with lack of progress in circular secondary material usage over the past decade as pointed out in the 2022 Country Report for Romania (*European Commission, 2022*). Moreover, its levels decreased, between 2015 and 2020, from 1.7% to 1.3%, while the EU average increased from 11.3% to 12.8% over the same period (*European Commission, 2022*). In terms of perceptions, a study elaborated by *Gherheș, Fărcașiu & Para (2022)* concluded that (i) "increased amounts of waste", especially as a consequence of "throwing garbage on the ground/in forbidden places" and "lack of recycling", nearby (ii) "air pollution", and (iii) "deforestation" are the three main environmental concerns in Romania. Therefore, the investigated national background does not represent a good example of plastic waste

management, even though the country has to attain very ambitious objectives in this regard until 2030.

In such a context, as observed from the literature, the complexity of strategies, directions of actions, best practices, and policies related to plastic waste management (PWM) requires (i) a multi-stakeholder approach, (ii) promoting and practicing engagement and behavioral change (*Huttunen et al., 2022*), (iii) a circular approach (*Nordic Council of Ministers, 2023*), (iv) asking for clear targets in terms of virgin plastic volumes, including fees for funding potential solutions across the plastic lifecycle, or more controlled disposal (*Nordic Council of Ministers, 2023*). (v) Adjustment to regional and local geographies and (vi) the necessity of acquiring community support for each of them might be added to the list of demands for a more efficient PWM, all called to be considered in practice.

This is one reason for calling for detailed studies that need to be implemented at all levels, including national, regional, and even local ones. Although discussions across scientific and gray literature, nearby reports, or other materials do exist, still more scientific research is required (*Coffin, 2023*) to respond to the plastic concern. Moreover, considering that research in Romania is generally not very extended, especially when referring to punctual analyses, the expectation is to identify a large gap in the literature devoted to Romanian circumstances related to plastic pollution. This research gap certainly contributes to the lack of efficiency in the PWM in this national context and the current work intends to analyze it, with great emphasis on future needed steps for filling this deficit.

Accordingly, our study aims to investigate the stage of scientific knowledge of one profound environmental concern in Romania, intending to reveal the role of linear economy prevalence and mismanagement practices in plastic pollution in this country while outlining downstream and upstream solutions to reduce it, including circular economy strategies. Besides the urgency, a lack of appropriate attention, with not sufficient orientation to analyzing, discussing, and understanding plastic pollution concern could be observed. Consequently, our study could prove useful for researchers, decision-makers, policymakers, local entrepreneurial actors, and educators, but also for common citizens interested in this stringent problem, with all its negative effects, different causes and sources, potential solutions for responding to it, *etc*.

## SURVEY METHODOLOGY

This work integrates two main approaches: (i) a bibliometric analysis feed by Web of Science and Scopus databases, carried out with VOSviewer, to reveal the current coverage of peer-reviewed literature related to plastic waste in Romania and (ii) the subject-based review to underline the main themes related to plastic waste management, plastic pollution of the natural environment and mitigating options in Romania in line with circular economy principles. In the latter case, other academic and non-academic sources are examined (*e.g.*, Google Scholar, mass media, policy articles, or environmental reports) to provide a comprehensive review and to outline the related knowledge gaps in Romania. In this way, we used search engines and cited those documents that proved to contain relevant information for plastic pollution, plastic waste management, or circular economy

topics in Romania (*e.g.*, plastic waste statistics or best practices). One limitation could come from the fact that other possible documents that are not available online through search engines may be overlooked.

Accordingly, the main aim of this study was to observe and analyze how the topic of plastic pollution was investigated in the Romanian context across the scientific literature, but also in different non-academic sources. Firstly, to obtain a general overview of the theoretical advancement of the current topic in this particular national context, a quantitative approach was applied. As a second step, the in-depth analysis rounded the first one, while offering the circumstances to find out and investigate in more detail the sub-topics of plastic pollution, namely macro and microplastic concerns. Therefore, the main points addressed in this article were divided into particularities related to these types of concerns, and nearby common and/or particular measures for mitigating them, as shown in the conceptual framework from Fig. 1.

In such a context, as a method currently growing in most disciplines, the bibliometric analysis was used, based on data from Web of Science (WoS) Core Collection (TS = Romania AND TS = Plastic) and Scopus ("Romania" AND "plastic") databases, without imposing any limitation in time. Accordingly, we first looked for a more general search, focused on the particular interest research field, *i.e.*, plastic in Romania. For this, searching for plastic within all fields, the Romanian context was included in this process as a keyword that must appear within the selected studies. In the case of Web of Science, 142 articles that appeared between 1975 and 2023 were found, with an observed slight increase in the number of publications starting from 2018. Moreover, the first study with interest in our topic, a conference article entitled "Waste Recycling in Romania, a Component of Waste Management Case Study—Economic Model for Evaluating a Recycling Process", appeared in 2008 (*Ucenic & Topalu, 2008*) followed by an article from 2015, entitled "The Plastic Materials Impact on Environment and Health. Population Awareness in Romania" (*Niculae et al., 2015*) and other conference article, "Study Regarding Waste Management in Romania" from the same year (*Street, 2015*). Moving on to Scopus, the search selected 240 articles published between 1963 and 2023 (21 articles between 1963 and 2000; 40 articles between 2001 and 2010; in 2021–16 articles; in 2022–16 articles; in 2023–17 articles). The most numerous ones were published in 2018 (23 articles). These results translate into the fact that this topic of research is a new one, under-investigated in the Romanian context, at least when analysing the studies published in WoS and Scopus. Still, if considering that these databases are the most important ones when discussing scientific research and dissemination of its results, the conclusion might be seen as a general one, with solid arguments for concentrating the efforts on this topic of great interest in our current world.

For a more detailed perspective, two types of procedures were used and they generated different term co-occurrence maps. The first one is based on keyword analysis. In this situation, when analysing the selected articles, all keywords (both authors' keywords and keywords plus) were included in the investigation. In each case, the minimum number of occurrences of a keyword was established. From the entire list containing keywords, some of them are assumed to meet the threshold. For each of these keywords, the total strength

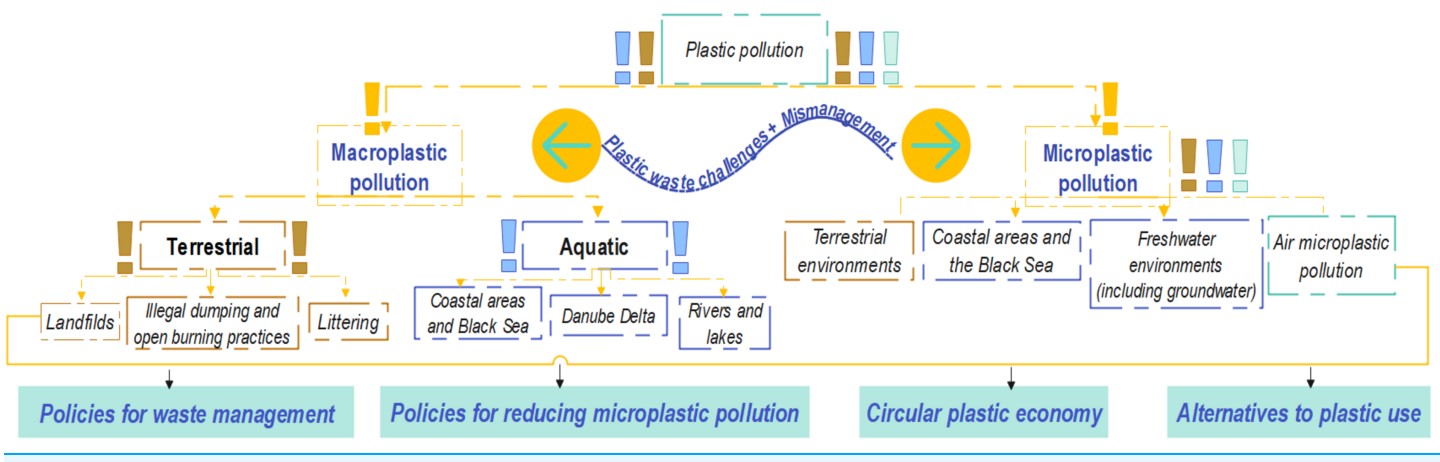

**Figure 1 Conceptual framework for addressing plastic pollution concern in Romania.**

of the co-occurrence links with other keywords was calculated. Then, the keywords with the greatest total link strength were selected. Moving on, the second type of analysis is based on terms co-occurrence from title and abstract. Of several terms, some meet the threshold of an established number of occurrences. For each of these terms, a relevance score was calculated and the most relevant were selected, with the default choice of selecting the 60% most relevant ones.

In addition, as observed in a second search, plastic pollution in the Romanian context is even a much less investigated theme, completing, in this way, the conclusion regarding the need for research in this direction (see details in the Supplemental info section).

## BIBLIOMETRIC ANALYSIS

As observed from the co-occurrence map based on keywords from the articles found in Web of Science (see Fig. 2A), three clusters with importance for our topic were obtained: (i) the red one containing terms like Romania, recycling, plastics, pollution, environment, waste; (ii) the green one integrating impact, behavior, determinants, plastic waste concepts; (iii) the violet one composed by management, waste management, energy, circular economy. It can also be noticed that the Romania keyword especially co-occurred with concepts like management, waste management, circular economy, recycling, plastics, environment, pollution, or determinants. The plastics keyword is observed to especially co-occure with management, recycling, plastics, environment, pollution, and behavior (see Fig. 3A), while the plastic waste keyword appears especially near terms such as management, circular economy, energy, determinants, behavior, and sustainability. Completing the perspective, the pollution keyword is closely linked to terms such as environment, risk, impact, waste, management, plastics, or Romania. In addition, underlying the novelty of the subject, at least in this particular national context, the topic of plastic waste, plastic pollution, circular economy, and the other keywords from the three clusters with interest for our research appear to be more recent, concentrated around the 2020 year, if compared to the studies from the medicine area (see Fig. 1A from Supplemental Information). Completing this analysis, the co-occurrences of the terms

based on text from titles and abstracts of the studies included in the investigation from Web of Science were also investigated. Thus, it was found that, among the five clusters formed in this type of analysis, the most relevant for our study are (i) the violet one with terms such as waste, plastic waste, waste management, glass, amount, and (ii) the red one containing concepts like process, model, research, value, sample, soil, water, effect, quality, solution (see Fig. 2A from Supplemental Information). In this analysis step, plastic waste co-occurred with terms like waste, waste management, amount, problem, effect, water, area, value, level, or case (see Fig. 3A from Supplemental Information).

In the same way, following a similar procedure, when generating the keyword co-occurrence map extracted from the Scopus database, although all clusters appear to be of interest for our study, (i) the one related to economics, waste management, waste disposal, or environmental protection, nearby (ii) the one strictly referring to plastic, plastic waste, microplastic, plastic pollution, Europe, and Black Sea, and (iii) other containing plastic recycling, packaging, recycling, soils, and European Union are the ones with major importance. These clusters appear to have integrated articles that were published around 2020 year, representing recent research approaches that reveal, even in this case, the trendiness of the topic investigated in this article (see Fig. 1B from Supplemental Information).

If comparing to the keywords co-occurrence map obtained from Web of Science articles, the map generated using the ones from Scopus is more complete and integrative in the case of our topic. Accordingly, terms like plastic pollution, microplastic, or plastic recycling appear in this map (see Fig. 2B). In this context, it should also be taken into consideration that the articles found in Scopus are considerably more numerous than the ones from Web of Science. In detail, the plastic waste keyword especially co-occurred with concepts like plastic recycling, packaging, waste disposal, waste management, Europe, European Union, Black Sea, plastic pollution, or microplastic. Moreover, the plastic pollution keyword co-occurred with terms such as plastic waste, waste management, waste disposal, environmental protection, microplastic, risk assessment, plastic recycling, Romania, or Europe (Fig. 3B). For a larger perspective, as in the case of the articles from Web of Science, the co-occurrences of the terms based on text from titles and abstracts from Scopus were also considered and analysed. In this case, only one cluster appears to be of interest for this article, namely the one containing plastic pollution, recycling, packaging, waste, environment, production, consumer, or need terms (see Fig. 2B from Supplemental Information). The keyword plastic pollution especially co-occurred with terms like environment, waste, management, packaging, product, recycling, region, interest, or awareness (see Fig. 3B from Supplemental Information).

As a synthesis, it can be mentioned that the most numerous studies identified both in Web of Science and in Scopus appear to analyse especially surface issues of plastic pollution, without concentrating on specific and detailed investigations of (mismanaged) macro and microplastics (*e.g.*, landfills, illegal dumping sites) in Romania. To respond to this concern, while completing the current bibliometric analysis with an in-depth exploration of the identified studies, main research gaps and potential future research directions to be approached are further discussed. This is made for a piece of better
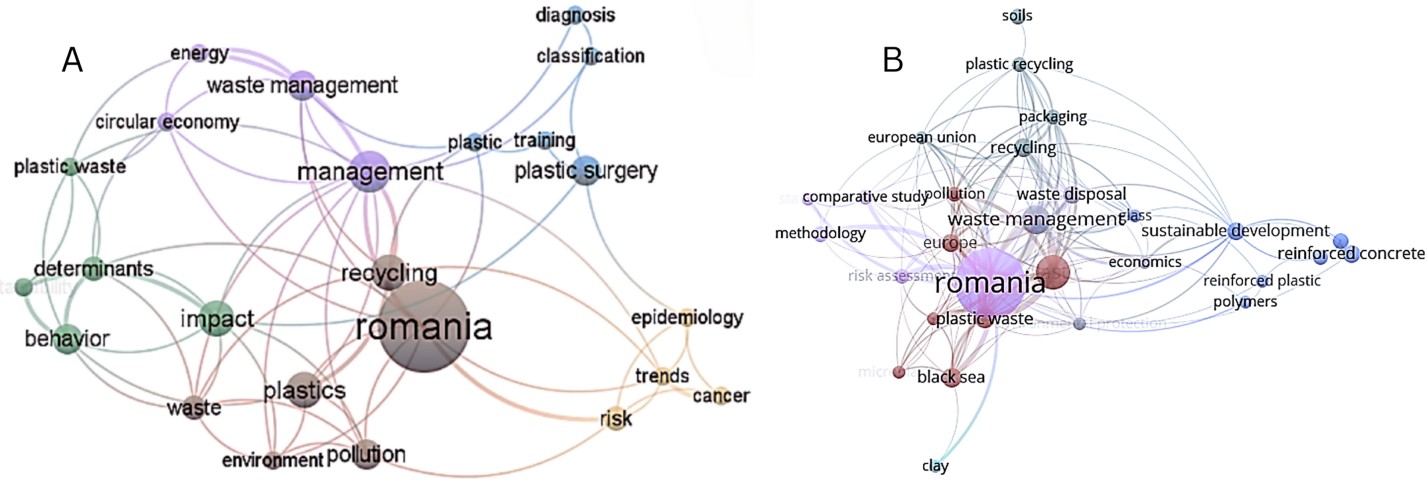

**Figure 2** **(A) Network visualization—WoS. (B) Network visualization—Scopus.** Note: Significance of the colors: different clusters formed. Figure created using VOSviewer.

knowledge on this particular topic in Romania, with an approximate complete overview of it, from the perspective of the synergies among different types of behaviours, environments, factors, *etc.*

# PLASTIC WASTE MANAGEMENT

## Plastic waste management challenges in Romania

Based on the data provided by municipalities in a general survey, the quantity of plastic waste collected *per capita* in 2008 was equal to 1.32 kg for towns and cities, 2.15 kg *per capita* in large cities, and 1.38 kg in rural areas (*Centru de Excelenta pentru Dezvoltare Durabila (CEDD), 2010*) while expansion of waste collection coverage in both urban and rural areas did not reach higher levels. Therefore, the uncollected household waste flow (with plastic waste materials included) ended up in terrestrial or aquatic environments in 2010–2018 while the delays in operating new sanitary landfills in some counties led to temporary dumpsites or transportation to municipal waste from one county to another on longer road routes (*Romanian Court of Accounts, 2022*). The supervision of the municipal waste management sector reveals that urban and rural municipalities besides the business sector are still linked to environmental crimes regarding the non-compliance of source separation of municipal waste, poor traceability of waste with significant repercussions to low recycling rates of plastic waste, and exposure to illegal waste trade activities (*National Environmental Guard (GNM), 2020*; *Mihai & Ulman, 2024*). These poor results in terms of waste collected amounts are driven, besides the low levels of coverage with efficient collection services (*Magrini, D'Addato & Bonoli, 2020*; *EEA, 2022*), by the low involvement of citizens, organizations, and institutions in the process. Accordingly, the necessity of educating and creating awareness across society regarding what is supposed to be one sustainable and especially pro-environmental behavior is frequently called in the literature dedicated to the current topic (*Golumbeanu et al., 2017*; *Tudor et al., 2022*). In this way, there is a need for an increased level of knowledge shared, including through educational
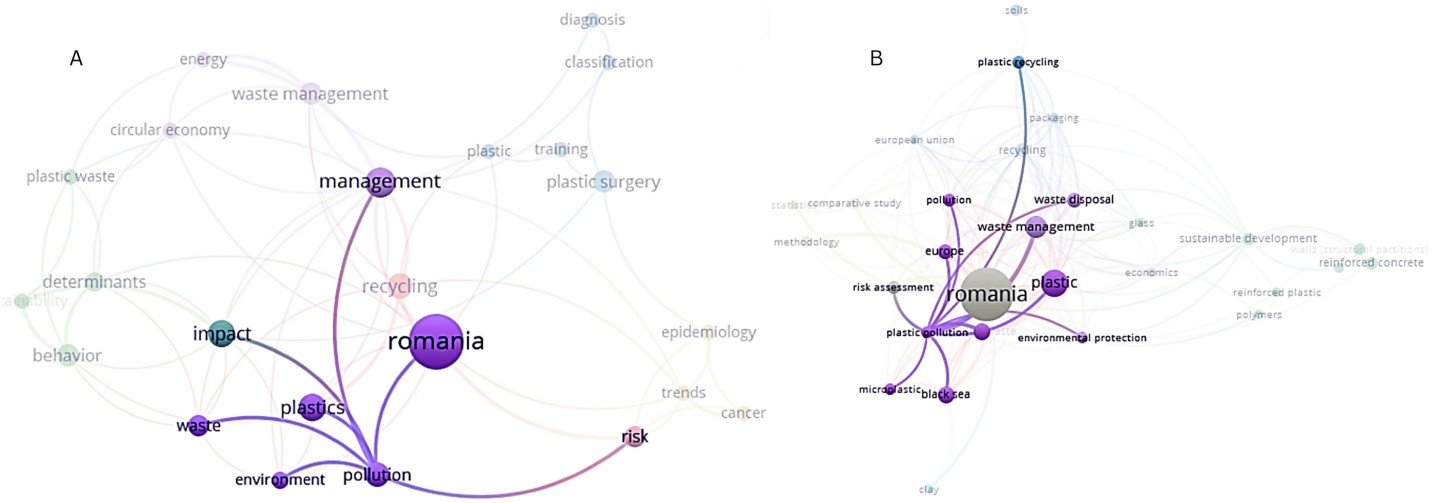

**Figure 3** (A) Network visualization with a focus on pollution—WoS. (B) Network visualization with a focus on plastic pollution–Scopus. Note: Significance of the colors: different clusters formed. Figure created using VOSviewer.

curriculum (*EEA, 2022*; *Geerken, Manoochehri & Di Francesco, 2022*). Its main roles would be related to consolidating positive attitudes towards environmental problems, strengthening collaboration, and changing patterns of behaviors (like today's throwaway culture) that feed plastic pollution in the natural environment. The Romanian decision-makers should attentively look at and try to follow the best practices provided by other European countries while adapting them to the national specificities. This adaptation need is especially supported by the observed resistance among consumers to different circular practices and an absolute preference for ownership in Romania, among other EU countries (*Piciu, 2019*; *Purcărea et al., 2022*).

## Mismanagement of plastic waste in Romania

Mismanagement of plastic waste (MPW) refers to unsound disposal practices that could feed the plastic pollution of the natural environment such as open dumping/illegal dumping practices, littering behavior, open burning of plastics, or disposal in non-sanitary landfill sites. These bad practices in Romania are related to poor logistics, particularly in rural regions (*Mihai, 2015*; *Mihai & Ulman, 2022*). Research studies regarding global plastic pollution levels use national data about the per-capita waste generation rates and plastic waste fraction of total municipal waste flow to determine the mismanagement of plastic waste (MPW%). For Romania, *Jambeck et al. (2015)* use an MPW rate of 28% while *Lebreton & Andrady (2019)* and *Borrelle et al. (2020)* use a MPW rate of 34%, the average result from these three studies being equal to 32%. The littering rate is assumed to be 2% at the global level (*Jambeck et al., 2015*) and this value is used for subsequent studies. *Law et al. (2020)* propose an MPW rate of 32.5 % resulting in 200,305 t of MPW in Romania. The per-capita mismanagement of plastic waste in 2019 was 2.69 kg in this country (*Ritchie, Samborska & Roser, 2023*), based on the study of *Meijer et al. (2021)* that points out that river plastic transport is still understudied while smaller rivers could significantly contribute to the plastic pollution of oceans. *Liro et al. (2023)* calculated the

MPW for all rivers from the Eco-Carpathian Region including Romania, Ukraine, Poland, Slovakia, Hungary, and the Czech Republic. The authors took into account the MPW levels using the baseline from *Lebreton & Andrady (2019)* but adapted to hydro-geomorphological conditions supported by a GIS analysis. However, these MPW levels must be refined with subnational data based on regional waste statistics including urban and rural gaps (*Mihai, 2018*) to improve the current plastic pollution models at regional scales.

# MACROPLASTIC POLLUTION IN TERRESTRIAL AND AQUATIC ENVIRONMENTS

## Macroplastic pollution of the terrestrial environment

### Landfills

The closure of non-compliant landfills is followed by the post-monitoring process to detect the risks of environmental pollution (*Sluser et al., 2017*). The most problematic issues in terms of plastic waste management in Romania were pointed out in its Circular Economy Strategy. Low recycling levels (31%), reduced content of recycled materials in new products, and low consumer awareness (*Ministry of Environment, 2022*) were especially indicated in this official document. As also emphasized in the literature, the norm in the last decades in Romania is given by the non-compliant landfills, with plastic waste that is known to tend to infiltrate into the soil and further into aquifers or reach nearby lands *via* wind (*Pop et al., 2021*). Microplastics tend to be released during these fire events at landfill sites (*Wojnowska-Baryła, Bernat & Zaborowska, 2022*) or by regular gas emissions, and carried out by the wind. Leachate is another important route for macro and microplastic contamination of soil, aquifers, or surface water bodies in the proximity as demonstrated in Serbia (*Narevski et al., 2021*). Besides mixed municipal waste flows, sewage sludge, or construction and demolition waste contaminated with plastics are still disposed of in landfills. Plastic waste accumulated could degrade in the landfill body as a source of microplastics (*Petrović et al., 2023*) but there are no such studies performed in Romania.

Therefore, future plastic pollution research should target the following topics regarding landfill sites: (i) leachate as the source of microplastics from closed or active landfills; (ii) former landfill sites–a source of plastic pollution for soil and groundwater; (iii) active landfill source of plastic pollution for surrounding areas.

### Illegal dumping and open burning practices

Illegal dumping of solid waste such as municipal waste, packaging waste, textile waste, e-waste, construction and demolition waste, and end-of-life vehicles including plastic items contaminate the terrestrial environments. Therefore, both natural and built environments are exposed to plastic pollution such as public lands, roadsides, forest areas, pastures, degraded lands, or floodplains (see Fig. 4). The presence of illegal dumping sites in certain geographical areas indicates that waste collection is not efficient, there is poor law enforcement, and low environmental awareness in the community.

The MPW flows are fed by domestic waste management deficiencies (*e.g.*, lack of proper waste collection schemes), nearby waste imports through illegal channels that contain

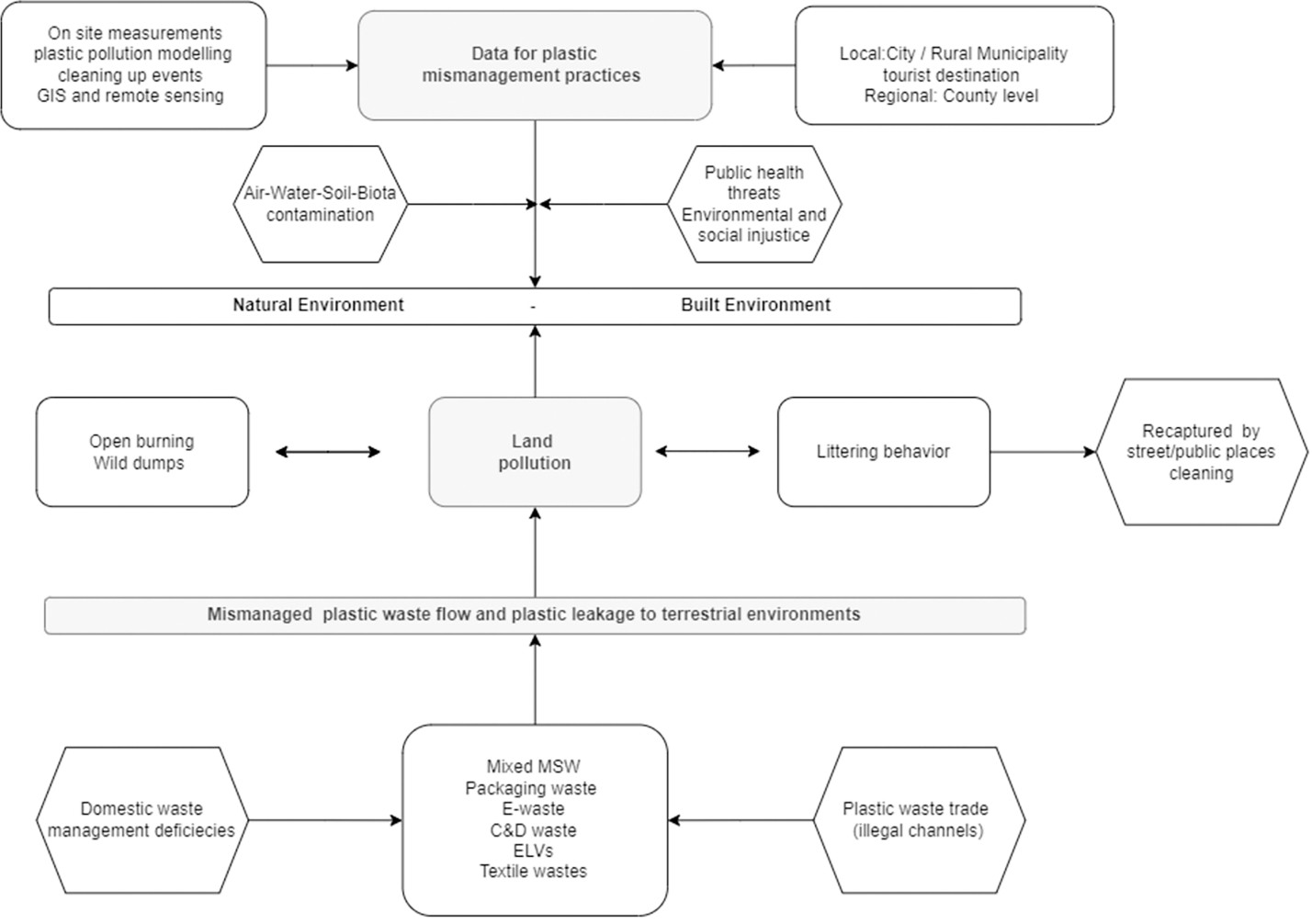

**Figure 4 Holistic analysis related to the opportunity to create a mismanagement plastic waste database in Romania.** Note: Significance of the *arrows*: interactions between plastic mismanagement practices, terrestrial environments, and the role of a database to examine this environmental threat.

non-valuable plastic items for the recycling sector such as e-waste and end-of-life vehicles. These end up in dumps, buried, or burnt while metal recovery *via* rudimentary practices such as copper and ferrous items are sold (*Modoi & Mihai, 2022*). A general survey of local authorities revealed that 536 communes had no access to waste collection services, favoring open dumping practices before 2010 (*Centru de Excelenta pentru Dezvoltare Durabila (CEDD), 2010*). The closure of rural dumpsites in 2009–2010 revealed that most of the localities from extra-Carpathian regions had the largest open dumps (surface and volumes). This is explained by the lack of poor waste management facilities, particularly in rural areas combined with higher population density (*Mihai, 2015*) while in the Carpathian regions, the dumps tended to be smaller and frequently located near freshwater bodies. In April-May 2019, a law enforcement campaign regarding illegal waste disposal practices in the proximity of water bodies was performed by the National Environmental Guard revealing 1908 dumps (C&DW, household waste, and other waste fractions) on

riverbanks while the total inventory of illegal dumping sites detected in open environments in Romania reached 3,837 sites covering 1,384,980 m$^2$ (*National Environmental Guard (GNM), 2020*). On this background, it can be assumed that the MPW near freshwater bodies tend to be mobilized by river flows and flood events in downstream areas (*Mihai, 2018*; *Liro et al., 2023*).

In the spring and autumn seasons, there is a traditional cleaning campaign of backyards and this practice could either eradicate some wild dumps with support from local authorities and NGOs or contribute to illegal plastic waste disposal or open burning practices as shown in Fig. 4.

Therefore, several actions are required to reduce MPW levels in terrestrial environments such as (i) GIS databases about the illegal dumping sites for both urban and rural regions; (ii) citizen-science projects in mitigating illegal dumping and open burning practices of plastic waste; (iii) better law enforcement with community support; (iv) cleaning up events; (v) source separation of waste improvement involving various stakeholders of the community.

## Littering

Littering is an intentional environmental crime behavior in disposing of plastic items on public lands in both urban and rural communities (streets, parks, parking areas, nearby shops, *etc.*) or during tourism and leisure activities (*e.g.*, beach, mountain trekking routes) contaminating the terrestrial environment. There are some cleaning-up events organized by local communities (*e.g.*, Let's Do It Romania) to divert such plastic waste from open environments. A comprehensive study of the littering behavior, magnitude, and composition was performed in 2009 by *Centru de Excelenta pentru Dezvoltare Durabila (CEDD) (2010)* for both urban and rural areas. PET bottles were the most littered bottle packaging material in rural areas (68.73%) followed by aluminum cans (16.67%). The findings of *Centru de Excelenta pentru Dezvoltare Durabila (CEDD) (2010)* are in line with later studies regarding PET bottles pollution of Bistrita River (*Mihai, 2018*) or cleaning up campaigns from recent years (*Mihai & Ulman, 2022*). As for food packaging materials, plastic waste dominated among littered items (39,53%) reaching higher levels in the case of bags (62.4%), while the overall plastic litter in rural areas was much higher (55.4%) compared to urban areas (12.2%) due to the lack of unsound waste management services (*Centru de Excelenta pentru Dezvoltare Durabila (CEDD), 2010*). In urban areas, cigarette butts were the most prevalent litter item in 2009 (*Centru de Excelenta pentru Dezvoltare Durabila (CEDD), 2010*) and persist nowadays as a bad practice among citizens (*Lakatos et al., 2023*).

In plastic pollution modeling studies, the litter rate is assumed to be 2% of the total MPW (*Lebreton & Andrady, 2019*) despite the uncertainties relating to this value.

There is no study to indicate the littering rate of MPW in Romania and an update about littering studies in urban and rural areas is needed to improve the plastic pollution modelling inputs.

### Macroplastic pollution of aquatic environments

Figure 5 reveals the current knowledge gaps and future research topics related to plastic pollution sources and loads of aquatic environments in Romania from direct inputs or transferred from terrestrial environments linked to solid waste and wastewater management deficiencies, and agricultural or industrial practices. Aquatic environments are represented by two major categories such as the Black Sea as a marine environment and the Danube River basin as the main freshwater environment. As observed from the bibliometric analysis, the Black Sea with its coastal areas and the Danube Delta were considered in the context of plastic pollution. It represents an advancement in the literature dedicated to this topic; however, these appear to be under-investigated from the distinction between macroplastic and microplastic point of view. Moreover, in the case of the Danube River basin, water body categories exposed to plastic pollution must be further investigated such as lakes (natural and anthropic), groundwater, rivers, and tributaries in diverse geographical conditions, and the Danube Delta. Special considerations should be given to the transboundary tributaries (*e.g.*, Tiza, Siret, Prut) that require international scientific cooperation to determine the magnitude of transboundary plastic loads from upper to lower sections of river systems toward the Danube River. This is also the case of this river course where upper sections are exposed to plastic pollution before reaching Romania (*Hohenblum & Maier, 2019*). On the other hand, the role of extreme events (floods, drought) in revealing the active plastic loads and historical pollution besides the plastic fragmentation and retention levels in river basins is essential to be quantified in Romania.

Furthermore, sectoral contributions to macro and microplastic pollution of aquatic environments related to municipalities (urban and rural), tourist destinations, agriculture and fishing activities, plastic industries, and transportation are largely unknown in Romania.

### Coastal areas and the Black Sea

The local population density and the differences in human activities, such as intense naval traffic, fishing activities like abandoned fishing gear (seines, trawl, purse, *etc.*) or even illegal fishing, nearby tourism activities and the Danube River, through the three discharge mouths are among the major factors that determine a high level of marine litter in the Romanian Black Sea area. These are frequently remembered across the existent studies (*Anton et al., 2013*; *Galatchi & Anton, 2020*; *Golumbeanu et al., 2017*). As a consequence, the waste abundance appears to be attributed especially to the direct anthropogenic influence (*Stoica et al., 2021*). *Golumbeanu et al. (2017)* clearly emphasized that plastic and related materials represent the most severe threat to the marine and coastal environment, being hardly degradable. Despite the local waste management efforts, plastic waste disposal near the marine environment (formed especially by bags, bottles, buckets, cans, linoleum, *etc.*) is seen as a macroplastic pollution source. It constitutes an aesthetic problem and also a threat to the biodiversity of the basin (*Galatchi & Anton, 2020*). In addition, tourism contributes to the plastic pollution problem through single-use plastic products (*Giurea et al., 2018*). In the same way, a kind of tendency among tourists to generate more waste
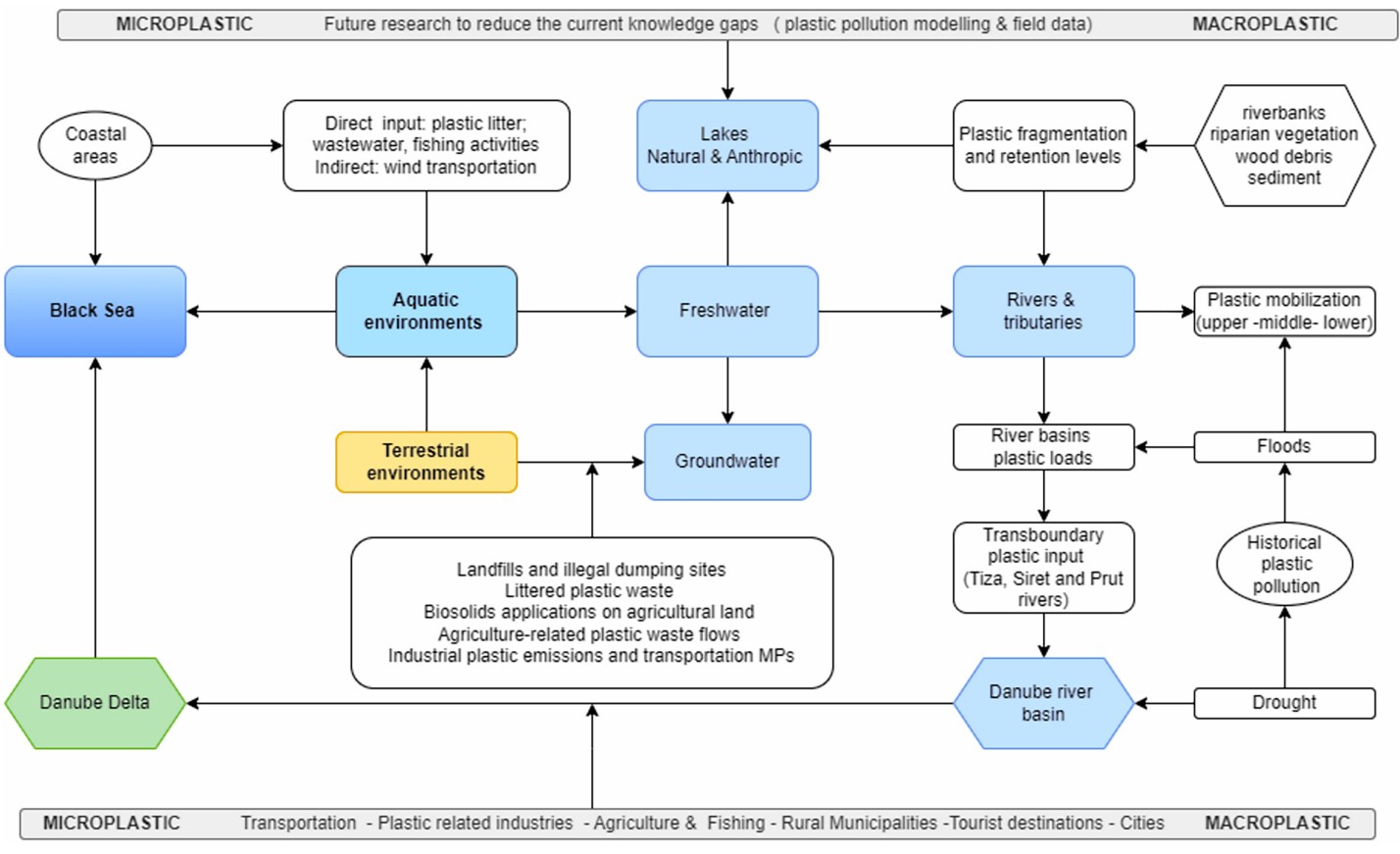

**Figure 5 Future plastic pollution research on aquatic environments in Romania to reduce the current knowledge gaps.** Note: Significance of the arrows: plastic waste emissions from land-based sources and interactions to aquatic environments (orange, terrestrial environment; blue, aquatic environments categories; green, biosphere reserve).

compared to local citizens was observed in other research (*Liu et al., 2022*). Still, detailed analyses in this regard in Romanian tourism destinations are lacking. In detail, according to Table 1, the ratio of artificial polymer is prevalent in all beach locations (>70%) and it ranges from 74.3% in Eforie to 91.9% in Vama Veche. Therefore, the contribution of tourist littering and fishing gear to plastic pollution of the marine environment is obvious, but the data broken down per sources (tourist *vs* fishing activities) is not available for beach litter.

Field measurements revealed that marine litter has a higher average density (81.5 item/km$^2$) compared to the Mediterranean Sea and plastic materials are abundant (*González-Fernández et al., 2022*). Still, the proximity of the Danube Delta could decrease this density to 30.9 ± 7.4 items/km$^2$ in the Romanian part of the Black Sea (*Suaria et al., 2015*). Therefore, regional differences are expected and the plastic contribution of the river network beyond the Danube River basin needs to be examined to determine the source of plastic pollution in the Black Sea region.

**Table 1 Artificial polymer ratio of total beach litter in Romania in 2021.**

| Beach | Artificial polymer (%) | Beach | Artificial polymer (%) |
|---|---|---|---|
| Corbu | 80.5 | Constanta | 83.7 |
| Navodari | 82.8 | Eforie | 74.3 |
| Mamaia Nord | 87.4 | Tuzla | 88.6 |
| Flora | 72.4 | Saturn | 80 |
| Malibu (h.c>) | 74.4 | Vama veche | 91.9 |

Note:
Data source credit: *Maritime Forum (n.d.)*.

### Danube Delta

The Danube Delta is a unique biosphere of Europe, a crucial wetland system and protected area, but it is exposed to plastic pollution. In terms of macroplastic, the domestic sources are related to Sulina towns and rural communities' mismanagement practices besides the additional plastic pollution leakage associated with tourism flows. However, there are no studies to reveal the magnitude of domestic plastic pollution leakage or tourist contribution to the plastic waste management problem in this area. The last environmental report of Biosphere Danube Delta does not include any mention in terms of plastic pollution of the aquatic environment. However, it specifies that there is an increase in plastic packaging materials (PET bottles and lightweight) in both urban and rural localities due to consumer goods (*Başcău, Ivanov & Niţu, 2022*). The municipal waste management is landfilled-based and some manual sorting for PET bottles is performed at Sulina (city-port), Chilia Veche, and Sfantu Gheorghe, while the separate collection of household waste increased a bit from 18.9% in 2021 to 21.5% in 2022 (*Başcău, Ivanov & Niţu, 2022*). Edighiol is more an isolated location and a higher percentage of artificial polymer could be attributed to fishing gear on the Black Sea. Vadu and Vadu Sud are wild beaches with tourists' contributions to beach litter through plastic packaging materials, but also paper/cardboard, aluminium cans, or glass bottles as shown in Table 2.

### Rivers

Plastic pollution modeling of the Bistrita river basin was performed by *Mihai (2018)* for the upstream sector of Izvoru Muntelui Lake and surrounding localities. This study tested the local river plastic pollution generated by villages with data obtained from sanitation campaigns (*e.g.*, Schitu creek, Ceahlau commune).

The first mapping of MPW along all Carpathian rivers is performed by *Liro et al. (2023)* while taking into account the entire Eco-Carpathian region where median values of MPW in Romania were the highest (118.4 tons.yr.km$^2$) along the fifth-order watercourses. This study relies on MPW data at the national level derived from *Lebreton & Andrady (2019)* providing a general framework of macroplastic potential magnitude. However, regional plastic pollution modeling is required using the local waste generation rates (urban and rural) and municipal waste composition data (*e.g.*, plastic waste fraction % of total MSW stream) to adjust the MPW levels to local geographies and waste management performances.

**Table 2 Beach litter included in the Danube Delta Biosphere protected area.**

| Beach | Artificial polymer (%) | Year |
|---|---|---|
| Edighiol | 93.9 | 2021 |
| Vadu | 86.3 | 2021 |
| Vadu Sud | 84.5 | 2021 |

**Note:**
Data source credit: *Maritime Forum (n.d.)*.

Plastic pollution in the Danube basin is significant for both macro and microplastic flows and international cooperation is crucial to reduce this environmental threat (*Miklos, 2022*). Transboundary plastic pollution is documented in the Tiza river for microplastic (*Kiss et al., 2021*) but further macroplastic studies are required in each country of the river basin (*Molnar & Hanko, 2022*). For example, in Romania, plastic pollution mobilization from the Mures and Somes rivers towards the Tisza river (Hungary) is largely unknown. Also, there is a lack of plastic pollution research in the case of other important transboundary rivers of the Danube basin such as Siret (Ukraine, Romania) and Prut rivers (Ukraine, Romania, Republic of Moldova), and how these river systems feed the plastic pollution of the lower Danube sector and delta region in Romania. Both Ukraine and the Republic of Moldova are facing waste mismanagement practices that contribute to plastic pollution of aquatic and terrestrial environments (*Procházková, Ivanova & Muntean, 2019*; *Safranov, Prykhodko & Mykhailenko, 2023*). Therefore, macroplastic pollution studies based on field measurements for different seasons and sections of rivers from the Siret and Prut basins are required for Ukraine, Moldova, and Romania. The river basin management plans must include data and analysis concerning plastic pollution levels.

### Lakes

*Mihai (2018)* tested the plastic pollution modeling generated by rural localities of the Bistrita River basin upstream of the Izvoru Muntelui with the data resulting from sanitation campaigns. The results showed that rural communities contribute to 85% of plastic pollution, the rest could be attributed to business, fishing, and tourism activities. The floods of 2005, 2008, and 2010 mobilized the mismanaged plastic waste upstream and accumulated on the lake water surface and shorelines in the mix with wood debris and other household waste fractions. In the period between 2000 and 2010, most rural localities lacked or had rather a poor waste collection infrastructure where source-separated collection of waste was marginal (*Mihai, 2018*). In these circumstances, the uncollected waste ended up in rural dumpsites or was burnt (*Mihai, 2015*).

The cleaning-up campaigns revealed that Izvorul Muntelei Lake is affected by macroplastic pollution, particularly by pet bottles despite being one of the most recyclable items. Accordingly, the Bicaz clean-up events organized in 2022 and 2023 revealed the historical plastic pollution of Izvoru Muntelui Lake. The drought led to the retreated water line of the lake and macroplastics could be collected from inaccessible places including the forest areas near shorelines, bays, and plastic trapped by riparian vegetation and sediment (*Liro et al., 2023*).

## Microplastic pollution

### Terrestrial environments (soil/land)

Even though most of the plastic-related studies on agricultural land are performed in Asia and Europe (*Sa'adu & Farsang, 2023*), in Romania, there are significant knowledge gaps regarding the concentrations of microplastics in soil compartments or various agricultural or other land-use ecosystems. Besides the degradation of macroplastics related to mismanagement practices in municipal and agricultural waste systems, sludge application is demonstrated as a key source of MPs contamination of soils in rural areas (*Van Den Berg et al., 2020*). Although plastic materials are omnipresent nowadays, research studies on different geographical areas are still incomplete including terrestrial environments. In Turkey, 41.5% of extracted plastics from 10 soil samples were microplastics and 16.3% were macroplastics related to single-use greenhouse cover and irrigation pipes (*Gündoğdu et al., 2022*). Similar studies are required to be performed in Romanian soils near large urban areas, industrial sites, towns, and rural municipalities.

*Tudor et al. (2019)* discussed the problem of microplastic contamination in the soil as a general review, but no experimental studies conducted so far in Romania have been performed. The problem of sewage sludge application as a source of microplastics is recognized (*Tudor et al., 2019*) and some alternatives are proposed to divert such waste streams from landfills or agriculture (*Purdea, Rusănescu & Tucureanu, 2019*). The use of pesticides on agricultural lands leads to plastic packaging generation that could leak into terrestrial environments. In Romania, there are no available waste statistics related to agri-plastics, and research on this topic is limited (*Tudor et al., 2022*). However, there are some experimental studies to provide biodegradable alternatives to the use of conventional studies (*Râpă et al., 2011*), but updated investigations are required.

### Coastal areas and the Black Sea

Macroplastic fragmentation and unmanaged plastic waste from terrestrial and riverine environments are the Black Sea region's main diffuse sources of MPs (*Strokal et al., 2022*).

In the case of the Eastern coast of the Black Sea in Turkey, more attention is paid to research on plastic pollution (*Bat & Öztekin, 2022*; *Terzi et al., 2022*), if compared with the western coast of the sea, which is related to the delta and the discharge of the Danube River into the sea. To fill this knowledge gap in Romania, *Pojar et al. (2021a)* collected 12 samples from surface waters with a neuston net. The results showed an average concentration of seven plastic particles/m³ of which fibers (~76%) had the highest concentration, followed by foils (~13%) and finally fragments (~11%). Their statistical analyses related to the concentration of plastic show that it was significantly higher near the mouth of the Danube River than in those four regions along the Romanian and Bulgarian coasts. This shows that the Danube River has a significant contribution to the concentration of plastic, *i.e.*, macro, micro, and nanoplastic in the western part of the Black Sea. The plastic accumulation in the Danube Delta and Black Sea coasts poses risks to both freshwater and marine life (*Buruiana, Ghisman & Obreja, 2022*). MPs having a long persistence in the environment are transported by the flow and deposited in the sedimentary systems. The abundance and morphology of the particles change due to the

flow rate and speed of the currents (*Pojar et al., 2021b*). *Stoica et al. (2021)* found that cigarette butts and pieces of plastic/styrofoam (2.5–50 cm) accounted for more than 50% along three Black Sea beaches. Significant differences between Mamaia resort and the protected wild areas of Vadu were found, revealing, in this way, the role of tourism flows in the plastic pollution problem. According to *Strokal et al. (2022)*, an increasing collection of plastic materials and reduced consumption will diminish MPs from the sea by 40% till 2050. Therefore, improvement of waste management performances along the Black Sea coast is imperative in Romania to reduce the magnitude of plastic leakage into marine environments. However, the plastic waste transport from the Danube River basin to the Black Sea will remain a key environmental threat to be managed in the following decades.

### Freshwater environments (including groundwater)

Plastic pollution in the Danube River basin requires a complex analysis involving transboundary pollution (*Balla et al., 2022*) in various rich ecosystems with international cooperation on this matter (*Kittner et al., 2022*). In Romania, there are serious knowledge gaps in research regarding MPs' presence in all types of freshwater environments. Previous studies examined the organic and inorganic pollutants including micropollutants of transboundary rivers like Siret (*Zait et al., 2022*) and Prut (*Moldovan et al., 2018*), but no investigations concerning MPs were performed. Multisectoral cooperation (businesses-academic-institutions-authorities-NGOs) in relevant plastic research projects is undertaken in Romania. Microplastics were found in all water samples collected from 10 rivers, four lakes, and the Danube River course in Romania, suggesting that microplastics represent widespread pollution in freshwater environments. Thus, further investigations are required to compare the results (*Act for Tomorrow, 2021*). The first comprehensive report on the magnitude of plastic pollution on the Danube River in the Romanian sector reveals the fact that the Danube-Black Sea river transport has an average of 48.5 tons of MPs and 48 tons of macroplastic. The highest transport yield was recorded in the station sampling of Moldova Veche with values between 93 and 100 tons of plastic material transported per year with a volume of MPs between 46 and 51 tons (*Mai Mult Verde Association, 2023*). In this collaborative research, water samples of surface water (0–0.6 m) and depth were performed. Their results reveal that MPs have higher concentrations related to the retention process of river rocks/minerals, fauna, and flora (*Mai Mult Verde Association, 2023*).

*Kiefer, Knoll & Fath (2023)* revealed a high concentration of MPs in the Danube River Delta with a maximum value of 2,677 p·L$^{-1}$ (>20 μm in size) and requested standardized sampling methods with different mesh sizes to avoid underestimation of MPs concentration. The problem of the harmonized approach of MPs in the Danube is also claimed by other studies (*Kittner et al., 2022*). Besides MPs transportation dynamics and retention levels by vegetation and sediment (*Pojar et al., 2021c*), the effects on freshwater biota are concerning and some species could be used as bioaccumulators of MPs pollution for freshwater ecosystems (*Stankovic et al., 2021*). Freshwater biota is exposed to microplastic pollution as shown in the case of *Chondrostoma nasus* with samples taken near the urban settlement of Mures river basin (*Curtean-Bănăduc et al., 2023*). Samples

collected upstream, downstream, and from nearby wastewater treatment plants on the Jiu River detected the presence of microplastics with higher organic load in the downstream sector (*Batrinescu et al., 2021*). In the case of the Dambovita River, the highest microplastic concentrations were detected downstream to the wastewater treatment plant with a significant difference between sediment (90 mg/kg) and water column (9 mg/L) (*Maria et al., 2023*). Laboratory tests were performed to detect microplastics of surface water *in vitro* conditions through the phthalates' desorption (*Scutariu et al., 2019*). According to *Nesterovschi et al. (2023)*, the presence of Mps in karst spring water was not reported until their study in Romania. Following the analyses, the presence of Mps was confirmed in the spring water samples collected from two karst springs in the Apuseni Mountains (Țarina and Josani), located in North-West Romania. They obtained a quantitative estimate expressed as the number of fragments or fibers per liter (0.034 at Josani and 0.06 at the Țarina karst spring in the spring of 2021 and, in the fall of the same year, 0.05 microplastics per liter). Most of the MPs found here were dominated by polyethylene terephthalate (PET), followed by polypropylene.

*Gundogdu et al. (2023)* highlighted the knowledge gaps related to MPs contamination in groundwater at the global level and their potential implications for both public health and the environment. Such studies need to be developed in Romania because it is an important source of drinking water. On the other hand, old landfills still represent an environmental threat to surrounding groundwater sources. Therefore, landfill sites could act as microplastic sources for groundwater (*Marinov & Marinov, 2014*) as well as for the application of sewage sludge in agriculture.

### Air microplastic pollution

Landfills and wild dumps fires that contain plastic items could release MPs into the air besides open burning practices of agricultural and household waste in rural regions. In addition, the burning of plastic waste from waste collection sites or recycling centers releases toxins into the atmosphere, smoking representing one of the main reasons in the latter case (*Dragan, 2021*). On the other hand, burning plastic waste in households releases toxic polycyclic aromatic hydrocarbons beside PM10 (*Hoffer et al., 2020*) while PVC burning poses concerns related to particles with cadmium content (*Kováts et al., 2022*), exposing rural populations to health risks. The burning of plastic waste is proven to be a source of microplastic pollution in Hungary (*Kováts et al., 2022*), while preliminary studies were performed for both Romania and Hungary countries (*Hoffer et al., 2023*). Thus, microplastic pollution research and investigations about the mismanagement practices such as open burning/illegal dumping of plastic waste, accidental fires at waste recycling sites, self-ignition (landfills), or intentional burning (waste collection points) as the source of MPs released in the air beside the role of wind transport in the case of lightweight plastic fragments from landfills, wild dumps, or sorting stations toward surrounding areas should be further considered.

The MPs research is gaining scope and public interest, according to *Pop et al. (2023)*, but it remains a research field that requires better geographical coverage, both at the European and worldwide levels.

## Policies to reduce macro and microplastic pollution in Romania under the circular economy framework

Figure 6 shows the holistic approach of reducing macro and microplastic pollution of terrestrial and aquatic environments in Romania. This involves the three following key areas: (i) downstream actions—recovering plastic from water bodies through floating barriers and automatic machines, cleaning urban water management infrastructures, anti-litter campaigns, and law enforcement for waste related crimes; (ii) plastic pollution database-including field data such as litter (from beach, river, marine, street, urban vs rural areas), on site waste disposal sites (landfills, illegal dumpsites), and plastic flows for all waste streams that contain plastics including from agriculture and fishing activities; (iii) upstream policies in line with circular economy ambitions—deposit return schemes and EPRs to increase quality of plastics for recycling industries and to curb the plastic imports in favor of domestic sources, expansion of reuse and refilling systems, reducing the plastic packaging material through other alternatives (such as glass, packaging with biomass feedstock), assuming the zero plastic waste approach at business level, and target cities and rural regions to zero-waste municipalities objectives.

Business actors are involved in the problem of macroplastic pollution of rivers. A Romanian start-up created an automatic river cleaning system that operated in Crisul Repede River (Bihor county) and this device aimed to collect over 500 tons of waste per year (Romanian Insider, 2019). The Floating HDPE Log Boom systems are already installed on five major rivers such as Ialomita, Dambovita, Arges, Jiu, and Mures (Cristea, 2022). The data collected from floating traps would help to further study the riverine plastic pollution transport features and their seasonality (van Emmerik et al., 2019). Authorities from the water management sector perform each year cleaning up events of lakes and rivers under their administration.

### Waste management

Improvement of waste collection efficiency is compulsory to reduce as much as possible the illegal dumping of household waste in the surroundings. The source separation of the dual system (dry and humid fractions) is inefficient and contaminates the plastic waste. The poor quality of plastic waste collected from domestic sources increases the refuse rate at sorting stations. The refused household waste (including plastic fraction) is usually disposed of in landfills. For each ton of refused waste, there is an additional fee of 80 lei (around 17 euros) as a contribution to the circular economy as in the case of landfilled waste. However, this fee is low and, consequently, favors the importation of plastic waste that ends up in Romania's landfills or natural environment (Mihai & Ulman, 2024). Gherheş, Fărcaşiu & Para (2022) indicated that, in the case of public institutions, the legislation regarding waste collection is coherent and comprehensible, but, when referring to households and blocks of flats, there are no clear methodological norms from the central authorities for a more efficient separate waste collection to be put into practice. The shift from a dual system to four or five fractions (paper/cardboard, plastic, metal, glass organics, residual) combined with strong environmental awareness campaigns could catalyze the recycling sector and plastic waste diversion from landfills and natural environments.

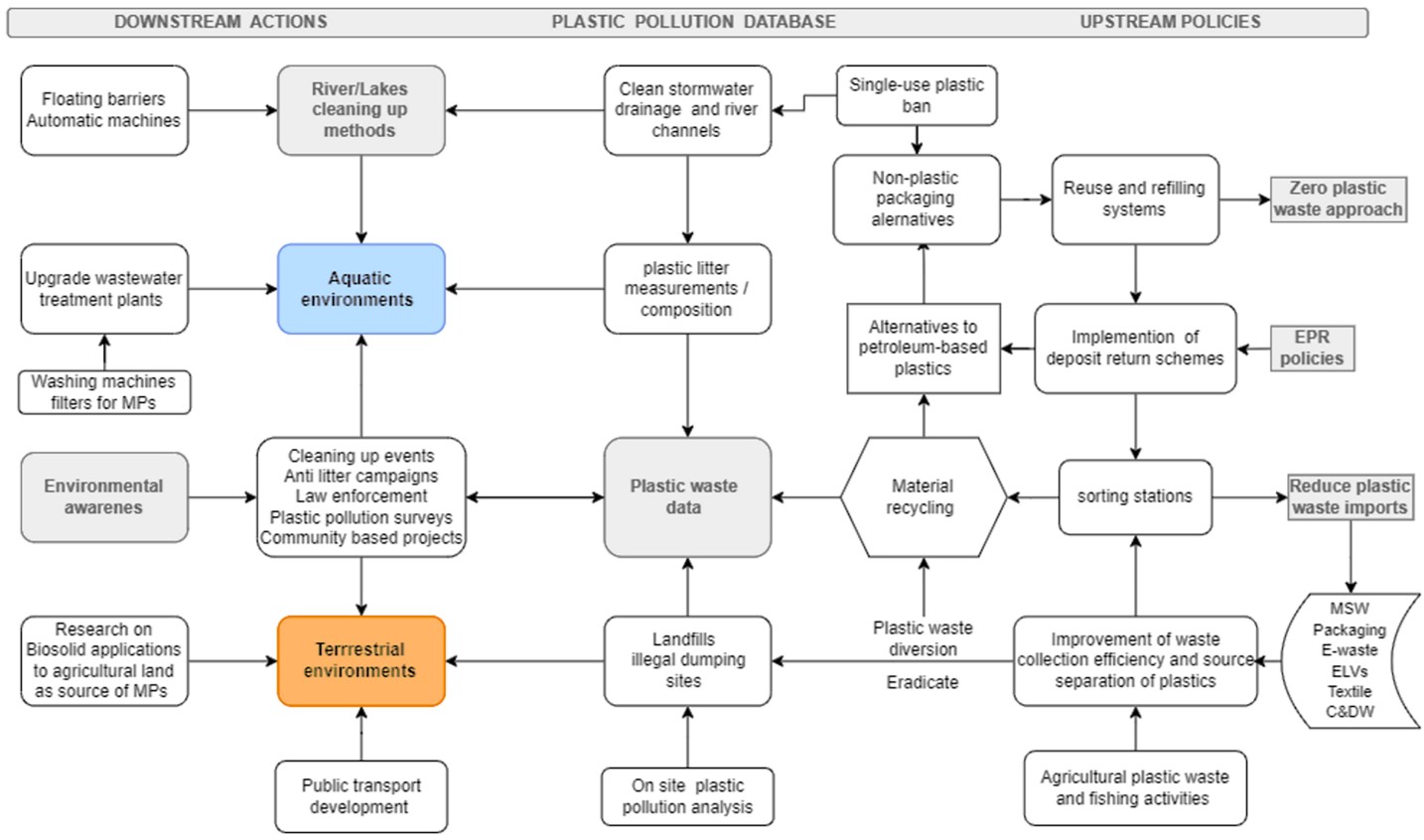

**Figure 6 Reducing plastic pollution and knowledge gaps through downstream actions and upstream policies under the circular economy framework.** Note: Significance of the *arrows*: interactions between plastic waste, environment, and key strategies to reduce this threat; Significance of the *colors*: blue color—aquatic environments, orange color—land/terrestrial environment, grey color—some key strategies to reduce plastic pollution and to adopt CE actions.

In this regard, the Salacea commune from Bihor County is a good example. Bihor County signed a partnership with Zero Waste Romania, in 2018, through which the concept of zero waste was implemented in the Salacea community, with benefits like saving money, faster progress, more support to sustainability targets, and waste diversion from landfills (*Zero Waste Europe, 2019*). In this way, the amount of waste was reduced in just 3 months and this result made Salacea to become the only community in Romania to fulfill the target of reducing the amount of waste in landfills (*Bodog et al., 2018*).

### Policies for reducing microplastic pollution

For a reduction of MPs pollution in Romania, as in other developing countries, a comprehensive approach with several ramifications is needed, involving primarily government policies, regulations in this direction of reducing pollution with MPs, public awareness campaigns (NGOs, questionnaires addressed to the population, mass media, *etc.*) as well as a close collaboration with all interested parties related to this type of risk. The major directions of action are composed of (i) upstream potential responses, namely the ones for microplastic pollution prevention or reduction of microplastic use, and emissions, and (ii) downstream ones, *i.e.*, those for microplastic pollution mitigation or

capture of microplastics, followed by their controlled disposal (*Munhoz et al., 2022*). However, the *European Commission (2023)* clearly emphasized the fact that the current legislation lacks a comprehensive approach. On this background, the reduction of MPs pollution in Romania requires a multi-sectoral approach, involving government policies, regulations, public awareness campaigns, local authorities' participation, and collaboration with plastic and recycling industries. Some potential policies that could reduce both macro and microplastic pollution in Romania are revealed in Table S1 (see Supplemental Information). It is essential to be careful and to recognize that the approach to MPs pollution requires a coordinated common effort of the Romanian government, the industry, the communities, and individuals. Periodic monitoring and policy adjustments based on new research results as well as technological advances can be essential for an effective long-term strategy that would lead to control and a significant decrease in the pollution of environmental factors in Romania.

### Circular plastic economy

An Action Plan for Implementing Circular Economy in Romania was elaborated by the authorities (*Romanian Ministry of Environment, Water and Forests, 2023*) with a major aim of increasing the capture rate levels of domestic plastic waste while following the circular economy approach. This plan aims to expand the rates of separate collection and waste recycling concentrated around pay as you throw system and the development of an adequate infrastructure for waste collection (*Mic & Mic-Soare, 2021*), but also with the necessity of increasing the pro-environmental education among Romanians, nearby innovation and collaboration between stakeholders with community support (*Modoi et al., 2022*). Plastic shopping bag bans were a type of policy option to mitigate single-plastic use at the consumer level (*Androniceanu & Drăgulănescu, 2016*). On the business level, effective recycling systems such as "bottle to bottle" promoted by Green Tech Buzau require high-quality source-separated "pet bottles". Because of high contamination rates derived from domestic source separation schemes, this company imports plastic waste from abroad to keep the recycling facilities in operation at optimum levels.

The implementation of a deposit-return scheme (DRS) in Romania could play a key role in providing a better quality of plastic waste products for recycling industries and decreasing their importation levels and downcycling practices but such system requires community involvement (*Cliza & Spătaru-Negură, 2021*). The new DRS system that started on 30 November 2023 is the key to fulfilling the material recycling rates related to plastic waste packaging materials. The recycling market is developed for PET and HDPE polymers, but other plastic waste materials are more difficult to recycle. The use of non-recyclable plastic products in cement factors as a substitute for fossil fuels raises environmental concerns and it should be avoided taking into account the waste hierarchy approach. Some laboratory-scale studies look for alternative reuse of plastic materials in the construction sector. For example, PET bottles could be used as concrete additives (*Baciu et al., 2022*) while polypropylene could be used in asphalt mixtures (*Buruiana et al., 2023*).

The recycling of HDPE for industrial-scale applications is required to improve the diversion rate of this plastic waste flow from landfills in Romania (*Teuşdea et al., 2020*).

The use of PVC plastic waste in producing ecological mortars is another alternative route for this plastic waste flow that is not well regarded in recycling markets such as pet bottles or HDPE packaging materials (*Aciu et al., 2018*).

## CONCLUSIONS

The topics of plastic waste and plastic pollution appeared to be under-investigated in the literature considering Romania as a case study according to the bibliometric analysis based on Scopus and WoS databases. These concentrated around the 2020 year and, in this way, the fact that such topics constitute mainly recent research approaches was revealed. Moreover, several knowledge gaps were identified and must be covered by future research such as (i) adjusting MPW levels to regional waste management performances and determining littering rates in urban and rural areas to improve the plastic pollution modeling inputs; (ii) examining plastic pollution associated with landfill/illegal dumping sites and waste imports; (iii) assessing sectoral contributions to macro and microplastic pollution of aquatic environments related to municipalities, tourist destinations, agriculture, *etc.*; (iv) determining retention levels of plastic in river basins and role of riparian vegetation; (v) analyzing microplastics presence in all types of freshwater environments and interlinkage between macroplastic fragmentation and microplastic; (vi) assessing plastic loads of transboundary rivers related to mismanagement practices; (vii) determining concentrations of microplastics in air, soil, and other land use ecosystems. Plastic pollution research relies on mathematical modeling and field data measurements while improvement of their data sources combined with the expansion of geographical coverage is required to refine future related research in Romania and beyond. The article argues some key directions with several specific actions under a holistic approach relevant to reducing plastic pollution threats in Romania (*e.g.*, freshwater cleaning-up methods, environmental awareness, plastic waste databases) and some upstream actions to catalyze the circular economy transition (*e.g.*, EPR policies, zero-waste approach, reducing plastic waste imports) that could be further implemented in Central and Eastern Europe, Southeastern Europe or other middle-income countries around the globe (*e.g.*, Asia, Latin America, Africa). International cooperation on particular plastic-related topics (*e.g.*, Black Sea Region, transboundary rivers, Danube river basin) is expected to emerge in the following years and multi-sectoral research is required to embrace and expand circular economy solutions in a landfilled-based country like Romania.

### Funding
This work was supported by a grant of the Ministry of Research, Innovation and Digitization, CNCS—UEFISCDI, project number PN-III-P1-1.1-TE-2021-0075, within PNCDI III. The funders had no role in study design, data collection and analysis, decision to publish, or preparation of the manuscript.

## Grant Disclosures

The following grant information was disclosed by the authors:

Ministry of Research, Innovation and Digitization, CNCS—UEFISCDI: PN-III-P1-1.1-TE-2021-0075, PNCDI III.

## Competing Interests

The authors declare that they have no competing interests.

## Author Contributions

- Florin-Constantin Mihai conceived and designed the experiments, performed the experiments, analyzed the data, prepared figures and/or tables, authored or reviewed drafts of the article, and approved the final draft.
- Simona-Roxana Ulman conceived and designed the experiments, performed the experiments, analyzed the data, prepared figures and/or tables, authored or reviewed drafts of the article, and approved the final draft.
- Valeria Pop conceived and designed the experiments, performed the experiments, analyzed the data, authored or reviewed drafts of the article, and approved the final draft.

## Data Availability

This is a literature review.

## Supplemental Information

Supplemental information for this article can be found online at http://dx.doi.org/10.7717/peerj.17546#supplemental-information.

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
