# Peer review of "Macro and microplastic pollution in Romania: addressing knowledge gaps and potential solutions under the circular economy framework"

_PeerJ, doi:10.7717/peerj.17546_

## Round 0.1 · original submission · Minor Revisions

We have received positive feedback, yet some minor changes are desired.

Reviewer 1 ·

Basic reporting

Needs Improvement

Experimental design

Appropriate

Validity of the findings

Satisfactory

Additional comments

This article is based on a bibliometric analysis feed by Web of Science and Scopus databases to reveal the current coverage of peer-reviewed literature on plastic waste in Romania. It also covers the subject-based review to underline the main themes related to plastic waste management, plastic pollution of the natural environment, and mitigating options in Romania that align with circular economy principles. The authors have identified several knowledge gaps that must be covered by future research to reveal the magnitude of both macro and microplastic pollution levels for each environmental compartment at various geographical scales.
Overall, the paper was written in an easy-to-understand manner. However, there are still some sections that need to be improved. Hence, this paper must be revised carefully before being considered for publication in a high-impact journal. I hope the comments below will help improve the paper further.
Throughout the manuscript, there are a lot of grammatical and syntax errors which need to be corrected.
Avoid using long sentences and break them into smaller ones to provide clarity. L76-83, L140-143, L232-237, L237-242

Specific comment:
Abstract:
- Major revisions must be made before the main content is amended.
- An abstract is often presented separately from the article, so it must be able to stand alone. Hence, the study's problem statement, aim, novelty and results should all be included in one paragraph of the abstract.
Which knowledge gaps did you find? Highlight them in the concluding sentences.

Introduction:
Rewrite the first paragraph to make it clear, concise, and meaningful.
The introduction should cover the gap in the research. However, it needs to be better covered in this section.
- Also, please mention the importance of this study to society and industry.
The problem statement of your introduction needs to be stronger; you need to discuss more about it.
- Revised Introduction section based on the structure below:
1st paragraph: Problem statement
2nd paragraph: Current ongoing solution
3rd paragraph: Proposed solution in this work.
4th paragraph: Summarized the current research novelty and objective of this work.

Materials and Methods:
- Please check the SI units throughout the manuscript.
- Please provide an additional figure to summarize the whole workflow.
- Please provide higher-resolution figures.

Conclusions
- Please include what was done in the study and the optimized results.
- Please include the limitations and what can be done in the future.

References
- Kindly revise the reference format according to the author's guidelines.
- Authors are encouraged to cite more recent and relevant literature from the target journal.

Reviewer 2 ·

Basic reporting

The scientific work is unambiguous
Literature references are adequate.
This article is of broad and cross-disciplinary interest and within the scope of the journal

Experimental design

The survey methodology is consistent and evidenced un adequate organization.

Validity of the findings

Conclusions are appropriately stated

Reviewer 3 ·

Basic reporting

This study identify the various sources and practices that generate macroplastic and microplastic waste across different terrestrial, marine, and aquatic ecosystems, as well as in the air. Additionally, identify the gaps in our understanding of these issues in Romania. The objective is also to highlight policies and actions that could promote a circular economy and reduce plastic waste. The methodology seem adequate, they perform a literature revision and a bibliometric analysis, to determine the actual knowledge of this topic in Romania. However, I think some details could be added to its methodology to clarify the way some information sources were identified.

Experimental design

In lines 102-104, it is mentioned that “other academic and non-academic sources are examined (e.g., Google Scholar, mass media, policy papers, or environmental reports) to provide a comprehensive review." It is crucial to specify how these information sources were located and selected. For example, in the case of environmental reports, how did you find them, and what criteria did you use to choose them? Did you visit government offices to request related information? Or in which databases are these environmental reports available? This level of detail is important for replicability and to determine whether there are other potential information sources that might have been overlooked, or whether the search for these various information sources was comprehensive.

Figures 2 and 3 are interesting, but the text is too small, making it hard to read. Please improve the figures to increase the font size for better visibility. Additionally, the legends need to be more descriptive—what do the colors represent, what does the node size signify, and what do the links indicate?

Figures 4, 5, and 6 are not very clear—they're hard to understand, and the figure legends do not provide enough information to guide the reader. What do the arrows represent? What do the colors mean? Consider using arrows of different colors to indicate various processes or effects. This could help make the figures more comprehensible. Additionally, a more detailed explanation in the legends would improve the clarity and usefulness of the figures.

Validity of the findings

I believe the validity of the findings is closely related to the search strategy, which appears to have been conducted thoroughly in two of the best databases in the world. My concern, however, is about how the other sources were identified and how the search was carried out, e.g. in Google Scholar. I assume the same keywords were used as in Scopus and Web of Science, but this should be clarified. I also think that the flowchart, typically used in systematic reviews, could be useful to illustrate the databases used and the steps taken. While I don't doubt that the authors conducted a thorough search, some specific details about this search process are lacking

Additional comments

I believe this research is interesting because it reviews the state of the art on a very important environmental issue: the disposal of macroplastics and microplastics into the environment. It also identifies potential policies and actions to promote a circular economy. The study uses Romania as an example of a European middle-income country, suggesting that some results could be applicable to other countries. Although I agree with this approach, I think the discussion and conclusion should address this point more directly and explain how these results can be applied to other countries worldwide.

---

## Round 0.2 · accepted · Accept

The authors have addressed all of the reviewers' comments!